# Ribosomes slide on lysine-encoding homopolymeric A stretches

**Kristin S Koutmou[1], Anthony P Schuller[1], Julie L Brunelle[1,2], Aditya Radhakrishnan[1], Sergej Djuranovic[3], Rachel Green[1,2]***

[1]Department of Molecular Biology and Genetics, Johns Hopkins School of Medicine, Baltimore, United States; [2]Howard Hughes Medical Institute, Johns Hopkins School of Medicine, Baltimore, United States; [3]Department of Cell Biology and Physiology, Washington University School of Medicine, St. Louis, United States

**Abstract** Protein output from synonymous codons is thought to be equivalent if appropriate tRNAs are sufficiently abundant. Here we show that mRNAs encoding iterated lysine codons, AAA or AAG, differentially impact protein synthesis: insertion of iterated AAA codons into an ORF diminishes protein expression more than insertion of synonymous AAG codons. Kinetic studies in *E. coli* reveal that differential protein production results from pausing on consecutive AAA-lysines followed by ribosome sliding on homopolymeric A sequence. Translation in a cell-free expression system demonstrates that diminished output from AAA-codon-containing reporters results from premature translation termination on out of frame stop codons following ribosome sliding. In eukaryotes, these premature termination events target the mRNAs for Nonsense-Mediated-Decay (NMD). The finding that ribosomes slide on homopolymeric A sequences explains bioinformatic analyses indicating that consecutive AAA codons are under-represented in gene-coding sequences. Ribosome 'sliding' represents an unexpected type of ribosome movement possible during translation.

*For correspondence: ragreen@jhmi.edu

## Introduction

Messenger RNA (mRNA) transcripts can contain errors that result in the production of incorrect protein products. Both bacterial and eukaryotic cells have evolved mechanisms to deal with such errors which involve (1) proteolytic degradation of the aberrant protein product, (2) mRNA decay and (3) ribosome rescue (*Shoemaker and Green, 2012*). One such mRNA surveillance pathway in eukaryotes targets mRNAs that lack stop codons (Non-Stop-Decay or NSD). In these cases, actively translating ribosomes are thought to read into the 3′ terminal poly(A) sequence of the mRNA triggering ribosome pausing as poly(lysine) is translated, followed by the recruitment of ubiquitin ligases, mRNA decay and ribosome recycling factors (review *Klauer and van Hoof, 2012*). Given the substantial amount of premature (or alternative) polyadenylation that has been documented in eukaryotes (*Ozsolak et al., 2010*), it seems that such an mRNA surveillance pathway might have considerable biological significance. Similarly, in bacteria, while no 'NSD-like' response has been characterized, it is known that poly(A) sequences are added to mRNAs in the process of being degraded (review *Dreyfus and Régnier, 2002*), and so ribosomes on these mRNAs may encounter similar challenges. The utilization in bacteria and eukaryotes of 3′ poly(A) tails as non-coding elements may reflect a common solution to the challenges for the ribosome in translating such sequences.

Most studies investigating how NSD works have been conducted in yeast using reporter constructs. Early studies in *Saccharomyces cerevisiae* revealed that mRNAs lacking stop-codons are targeted for decay both in a reaction dependent on the exosome-associated factor Ski7 (*van Hoof et al., 2002*) and in a more canonical degradation reaction involving decapping and 5′ to 3′

**eLife digest** Genes provide the instructions to assemble proteins from smaller molecules called amino acids. When a gene is 'switched on', the DNA that makes up the gene is copied into messenger ribonucleic acid (or mRNA) molecules, composed of building blocks called nucleotides. There are four types of nucleotides in mRNA molecules—commonly referred to as A, C, G, and U—and a set of three nucleotides is called a codon.

A molecular machine called a ribosome moves along an mRNA molecule translating the codons into protein. Each codon instructs the ribosome to add a particular amino acid to the chain of amino acids that will make up the protein. Some codons do not specify an amino acid but instead mark the point on the mRNA that the ribosome should stop and release the new protein. Most mRNAs have nucleotides beyond the 'stop' codon and these often contain a long stretch of A nucleotides, one after the other, which is known as the poly(A) tail.

Some mRNA copies may contain poly(A) tails before a stop codon, which can lead to the production of alternate and potentially harmful proteins. Cells have developed ways to identify and dispose of these mRNAs and their protein products. For example, in yeast and other eukaryotes, if an mRNA is missing a stop codon, the ribosomes will continue to translate along the mRNA into the poly(A) tail where they stall and are eventually removed. When this happens, the mRNA and protein are rapidly destroyed. However, it is not clear how this works.

Koutmou et al. studied the translation of a series of artificial mRNAs that contained different numbers of A nucleotides in codons of either AAA or AAG. Both of these codons specify the same amino acid, and should therefore be translated equivalently. The experiments show that the ribosomes read the AAA and AAG codons differently. When consecutive AAA codons are found in the mRNA, the level of protein production is significantly lower than when the mRNA contains iterated AAGs instead.

Koutmou et al. found that when ribosomes encounter consecutive AAA codons they undergo an unusual 'sliding' movement and are unable to accurately produce proteins. When a cell detects this abnormal sliding behavior, it rapidly triggers the destruction of the mRNA molecule. In contrast, when ribosomes encounter consecutive AAG codons, they slow down but do not slide, and therefore produce a correct protein.

Koutmou et al.'s findings also provide an explanation for why there are relatively few AAA codons within the regions of genes that encode proteins. The prevalence of alternative forms of mRNAs with poly(A) sequences before their stop codons suggests that ribosome sliding may contribute to an important pathway to control the activity of genes.

exonucleolytic degradation (*Frischmeyer et al., 2002*). Other factors involved in NSD have since been discovered; these include Dom34 and Hbs1 which facilitate ribosome rescue during NSD (*Izawa et al., 2012*; *Tsuboi et al., 2012*), Ltn1 and Not4 which ubiquitinate the protein products on non-stop mRNAs (*Dimitrova et al., 2009*; *Bengtson and Joazeiro, 2010*), and a number of other factors genetically identified as critical for poly(basic)-mediated stalling (*Kuroha et al., 2010*; *Brandman et al., 2012*; *Chiabudini et al., 2014*). Although many players in NSD have been identified and their functions defined, there remain critical gaps in our understanding.

In this manuscript, we focus on what must be the earliest events in NSD, the translation of poly (lysine) sequences by the ribosome. NSD is widely thought to be triggered by unfavorable electrostatic interactions that occur in the ribosomal exit tunnel when ribosomes translate the poly (lysine) sequences encoded by poly(A). Indeed, biochemical studies in rabbit reticulocyte lysate with proteins interrupted by iterated poly(lysine) and poly(arginine) sequences indicate that positively charged residues do slow translation and produce transiently arrested species (*Lu and Deutsch, 2008*). Other examples of peptide-mediated stalling have also been documented in bacterial and eukaryotic systems. In some cases, such as the *tnaC* gene, *secM*, or *ermCL* in bacteria, the peptide stalling motif is several amino acids in length and appears to specifically engage the contours of the exit tunnel to elicit stalling (*Gong and Yanofsky, 2002*; *Nakatogawa and Ito, 2002*; *Vazquez-Laslop et al., 2008*; *Seidelt et al., 2009*; *Bhushan et al., 2011*; *Ito and Chiba, 2013*; *Arenz et al., 2014*). Poly(proline) sequences have recently been shown to cause stalling during translation in bacteria and

eukaryotes in the absence of specialized bypass factors, EFP and eIF5A, respectively (*Doerfel et al., 2013*; *Gutierrez et al., 2013*; *Ude et al., 2013*). In this case, proline is thought to adopt a conformation that interferes with the ribosome active site geometry.

Here we take a high-resolution biochemical look at the molecular events that occur when the ribosome translates poly(lysine) peptides. We find that insertion of consecutive AAA lysine codons into reporters has a stronger negative impact on protein expression than insertion of an equivalent number of AAG lysine codons in both eukaryotes and bacteria. Kinetic and toeprinting studies in an in vitro reconstituted *Escherichia coli* translation system reveal that differential protein output is the downstream consequence of ribosome pausing followed by an unanticipated ribosome movement on successive AAA codons that we refer to as 'sliding'. When sliding occurs in the middle of genuine ORFs in a cell, frame is lost and ribosomes encounter out of frame stop codons that result in canonical (stop-codon mediated) termination. In eukaryotes, such premature termination events target the mRNA for non-sense mediated decay (NMD). The finding that the ribosome can robustly slide on poly (A) sequences explains bioinformatic analyses revealing that consecutive AAA codons are under-represented in ORFs in all genomes (unpublished data) and helps to rationalize the widespread usage of poly(A) sequence as a regulatory rather than a coding feature.

## Results

### Protein production is differentially diminished by iterated lysine codons (AAA vs AAG)

To begin investigating the translation of poly(lysine)-encoding sequences, we created a series of mCherry- and luciferase-based reporter constructs (*Figure 1A*) containing no insert, glutamic acid (GAA) repeats, or consecutive lysine residues encoded by various combinations of AAA and AAG codons. These reporters were introduced into *S. cerevisiae* and *E. coli* cells and the protein products visualized by either luminescence or fluorescence, respectively (*Figure 1B*). The insertion of twelve consecutive negatively charged glutamic acid residues (GAA) had no negative impact on production of the reporter protein (*Figure 1B*). By contrast, the addition of consecutive lysine residues generally resulted in overall less protein production (*Figure 1B*), consistent with previous studies of poly(lysine)-containing reporters (*Ito-Harashima et al., 2007*; *Lu and Deutsch, 2008*; *Chiabudini et al., 2012*). Interestingly, we find that protein output from the poly(lysine)-containing reporters is codon dependent in both bacteria and yeast; reporters containing iterated AAG lysine codons generate more protein than those with an equivalent number of synonymous AAA codons (*Figure 1B*). The relative differences in expression of AAG- vs AAA-encoded poly(lysine)-containing reporters in *E. coli* and *S. cerevisiae* are comparable ($4 \pm 0.3$-fold more in *E. coli* and $3 \pm 1$-fold more in *S. cerevisiae* from reporters with $AAG_{12}$ vs $AAA_{12}$).

### Kinetic analysis of lysine incorporation on consecutive AAA and AAG codons

One potential explanation for the codon-dependent expression of poly(lysine)-containing proteins could be that the ribosome more rapidly incorporates lysine on AAG than AAA codons. In *E. coli* a single tRNA with a UUU anti-codon decodes both lysine codons (*Chan and Lowe, 2009*), making *E. coli* an excellent system for studying differences in the production of poly(lysine) peptides. We measured the rate of lysine incorporation using a previously described reconstituted *E. coli* translation system (*Youngman et al., 2004*; *Gromadski et al., 2006*; *Zaher and Green, 2009*) on a series of lysine-encoding simple mRNAs including: AUG-AAA-UUC-AAG-UAA (MKFK-Stop), AUG-UUC-AAA (MFK), AUG-(AAA or AAG)5-UAA (MK(A or G)5-Stop). Only Lys-tRNA$^{Lys}$ was included during the translation of MKFK-Stop and MK5-Stop mRNAs while both Lys-tRNA$^{Lys}$ and Phe-tRNA$^{Phe}$ were present when MFK was translated. Electrophoretic TLC (eTLC) readily resolved the reaction products allowing for analysis of intermediate and complete peptide products (*Figure 2A*). The quantitated data were modeled in Mathematica using the kinetic scheme displayed in *Figure 2B* (see 'Material and methods'). These experiments reveal that addition of a single lysine in a heteropolymeric sequence is rapid and independent of whether lysine is the first or second amino acid incorporated (*Figure 2C*, rate constants for formation of MK and MFK peptides are 12 s$^{-1}$ and 7 s$^{-1}$, respectively); these rates are similar to those typically measured for peptide bond formation in this in vitro system (*Gromadski et al., 2006*). For messages containing iterated lysine codons, the rate constant for translating the first lysine codon is similarly fast ($k_{1,obs}$ from 2–19 s$^{-1}$, *Figure 2C*) for AAA and AAG

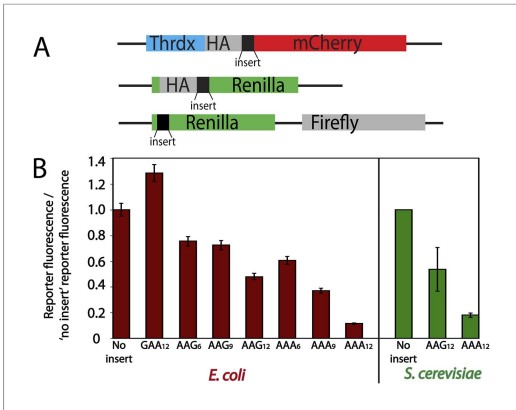

**Figure 1**. Protein production is differentially diminished by iterated lysine codons (AAA vs AAG) in *E. coli* and *S. cerevisiae*. (**A**) Schematics of the mCherry (top) and luciferase (middle, and bottom) reporters used in this study. The mCherry reporter contains an N-terminal thioredoxin (Thrdx) domain, 3HA-tag, sequence of interest (black section), followed by the C-terminal mCherry sequence. The top luciferase reporter includes a 2HA tag followed by sequences of interest (used for study in *Figure 1B*). The second luciferase reporter (used in *Figure 6*) has sequences of interest at the N-terminal end of Renilla. Firefly is used in this construct as an internal control in the second luciferase reporter. (**B**) Relative amounts of protein expressed from reporters expressed in *E. coli* (mCherry, red) and *S. cerevisiae* (luciferase, green). Error bars results from for the standard error of at least three experiments.

codons. However, subsequent lysines in an iterated sequence are added with considerably slower kinetics on both AAA ($k_{2,\text{obs}} = 0.0005$ and $k_{3,\text{obs}} = 0.0003$ s$^{-1}$) and AAG codons ($k_{2,\text{obs}} = 0.009$ and $k_{3,\text{obs}} = 0.015$ s$^{-1}$) (*Figure 2C*). We note that the rate of second lysine addition during the translation of MK$_5$-STOP messages are somewhat slower on AAA relative to AAG codons, potentially partially explaining the decreased overall protein output on these mRNAs. More importantly, however, these data show that the reactivity of the second Lys-tRNA$^{\text{Lys}}$ on iterated lysine containing messages (such as MK$_5$-Stop) is substantially reduced (at least 130-fold) on both lysine codon-containing mRNAs relative to normal elongation rates. Interestingly, the addition of a second lysine to messages with fewer sequential lysine codons (such as MK$_2$F-STOP) does not exhibit such a striking kinetic defect ($k_{2,\text{obs}}$ is not largely affected, data not shown). These data suggest that the identity of the message (i.e. a long poly(A) sequence) plays a critical role in the observed slowing of elongation. Toeprinting assays performed using the *E. coli* PURE cell-free translation system are consistent with these observations; *E. coli* ribosomes stall when the second lysine codon in iterated (AAA)- and (AAG)-codon containing sequences is positioned in the A site (*Figure 2—figure supplement 1*). Together, these results reveal that translating consecutive lysines in a poly(lysine) peptide sequence, either on iterated AAA or AAG codons, can lead to substantial kinetic delays in vitro.

## *E. coli* ribosomes add extra lysines on iterated AAA-containing mRNAs

As we explored the kinetics of lysine incorporation, we evaluated the ability of the ribosome to translate a variety of MK$_{(A\text{ or }G)2}$ di-lysine messages (*Figure 3A*). Unexpectedly, we found that messages containing iterated AAA codons generate extended peptides longer than the designed coding sequence (*Figure 3A*). When *E. coli* initiation complexes (programmed with fMet-tRNA$^{\text{fMet}}$) are reacted with Lys-tRNA$^{\text{Lys}}$ on messages containing two consecutive lysine codons followed by a variety of non-lysine codons (Phe (UUC), Val (GUC), or Stop (UAA)), only MKK peptide should be synthesized. However, we see the formation of a majority population of extended peptide product containing at least four lysines on all messages with two consecutive AAA codons (*Figure 3B*, lanes 2-4). In contrast, equivalent messages with two AAG codons predominantly form the expected MKK product (*Figure 3B*, compare lane 3 vs 5). We also find that a mixed sequence of lysine codons (AAA-AAG) can form some extended peptide (*Figure 3—figure supplement 1*). These data suggest that 5 As in a row are sufficient to promote the addition of extra lysines in vitro. We note that the identity of the codon that follows the di-lysine sequence is not relevant to the observed amount of extended peptide product (*Figure 3B*, *Figure 3—figure supplement 2*).

The production of peptide products containing more than the encoded number of lysines is surprising, especially given that there are no nearby upstream or downstream in-frame or out-of-frame lysine codons in these mRNAs (*Figure 3A*). We speculate that these extended peptides result from the ribosome repeatedly moving backwards by at least three nucleotides to position an AAA Lys codon in the A site, and then subsequent standard peptide bond formation. Toeprinting assays performed on iterated AAA- and AAG-containing mRNAs provide further support for such irregular movement of ribosomes specifically on iterated AAA codons (*Figure 2—figure supplement 1*); the toeprint on the iterated AAA sequence is diffuse relative to the discrete toeprint seen on iterated AAG sequence.

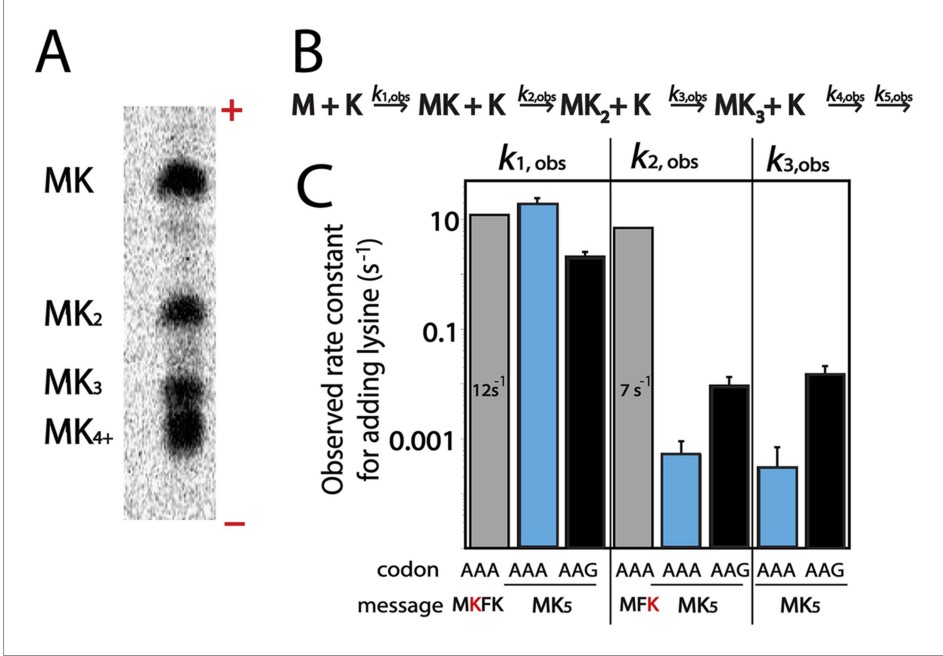

**Figure 2.** Kinetic defect observed on addition of second and third lysine residues in iterated lysine stretch. (**A**) Example eTLC displaying the *E. coli* translation products of a AUG-(AAA)$_5$-UAA message. The ± poles of the electrophoretic TLC are indicated. MK$_4$ and MK$_5$ products (and those with greater numbers of lysine) are difficult to resolve in this system but the other products are easily visualized. (**B**) Kinetic scheme for rate constants of sequential lysine additions to peptide chain. (**C**) Bar graph displaying rate constants for the addition of individual lysines to a variety of messages: MKFK-Stop (gray), MK$_{A5}$-Stop (blue), MK$_{G5}$-Stop (black), and MFK (gray).

The following figure supplements are available for figure 2:

**Figure supplement 1**. Ribosomes stall while adding a second lysine.

**Figure supplement 2**. Modeling of rate constants in Mathematica.

In the course of performing our experiments we carefully considered reports suggesting that T7 RNA polymerase could promiscuously add extra adenosines to poly(A) messages (*Tsuchihashi and Brown, 1992*; *Ratinier et al., 2008*); no experiment that we performed revealed any evidence for such heterogeneity in our mRNA products (*Figure 3—figure supplement 3*). Unlike better studied −1 and +1 frameshifting events, these data suggest that ribosomes on iterated AAA sequences are making unexpected and large excursions from their initial frame; we refer to this process as 'ribosome sliding'.

## Ribosome sliding is slow relative to the rate of normal elongation and termination reactions

The observation of ribosome sliding on iterated AAA codons is surprising given that the ribosome must somewhat regularly translate mRNA sequences in vivo that contain two consecutive AAA codons. While three or more AAA codons in a row are selected against in gene coding sequences, there are thousands of examples of two consecutive AAA codons in *S. cerevisiae* and *E. coli* genes (see further details in bioinformatic analysis below, *Table 1*). In the experiments described in *Figure 3A*, ribosome initiation complexes formed on the specified MK$_{A2}$-Stop and MK$_{A2}$F-Stop messages (*Figure 3B*) were only supplied with Lys-tRNA$^{Lys}$ and essential elongation factors; the subsequent substrates normally present in vivo after the formation of MKK peptide were left out. To determine if ribosome sliding occurs in more typical circumstances, we performed elongation reactions on the same mRNAs, but where both Lys-tRNA$^{Lys}$ and the relevant other downstream substrates (release factor 1 (RF1) or Phe-tRNA$^{Phe}$) were added to the ribosome initiation complexes.

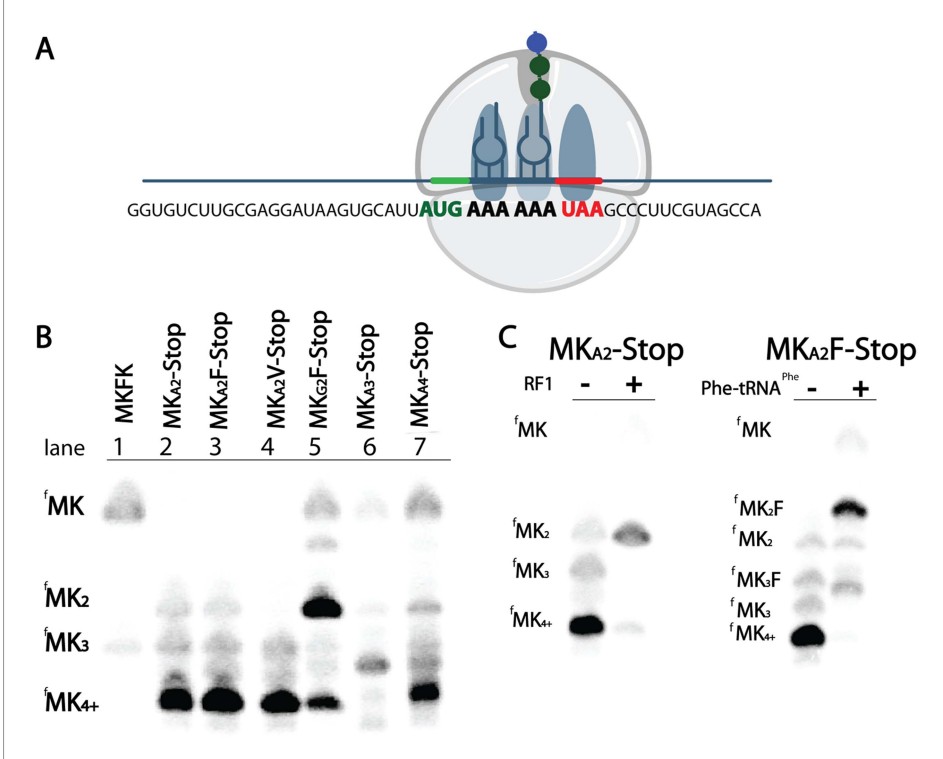

**Figure 3**. *E. coli* ribosomes add extra lysines on messages containing two sequential AAA, but not AAG, lysine codons. (**A**) Illustration of the ribosome on the entire MK$_{A2}$-Stop message. (**B**) eTLCs showing the peptide products resulting from translation of indicated messages with Lys-tRNA$^{lys}$ (but no other tRNAs or release factors) present. (**C**) eTLC displaying the peptide products resulting from the translation of indicated messages in the presence of Lys-tRNA$^{Lys}$ alone, or in the presence of Lys-tRNA$^{Lys}$ + factors (either RF1 or Phe-tRNA$^{Phe}$) necessary for messages to be fully translated.

The following figure supplements are available for figure 3:

**Figure supplement 1**. *E. coli* ribosomes add extra lysines to peptides translated on messages containing sequential AAA-AAG lysine codons.

**Figure supplement 2**. Quantification of the percentage of translated peptide containing more lysine residues than expected.

**Figure supplement 3**. T7 transcribed messages visualized on 15% denaturing PAGE gel.

The result is clear; in this latter case, the anticipated MKKF or MKK peptide products are predominantly generated (*Figure 3C*, *Figure 3—figure supplement 2*). These data suggest that ribosome sliding on iterated AAA sequences occurs more slowly than the normal rate of peptidyl transfer with Phe-tRNA$^{Phe}$ or RF1-catalyzed peptide release, respectively. Moreover, these results readily explain how the ribosome can normally translate (at least two) sequential AAA codons in vivo without sliding. When there are more than two AAA codons in a row, each lysine after the first is added slowly (*Figure 2B*), raising the possibility that sliding may become relevant on such messages.

## Ribosomes slide on poly(A)-containing reporters in an *E. coli* cell-free translation system

The initial in vivo observation that protein production is more severely impacted by iterated AAA than AAG codons (*Figure 1*) was recapitulated using the PURExpress *E. coli* cell-free translation system (NEB) (*Figure 4A*). This system contains all factors required for normal translation, but lacks cellular

**Table 1**. Bioinformatic analyses of poly(lysine) sequences

| Organism | Sequence | Occurances | Fraction observed | Fraction expected | Enrichment |
|---|---|---|---|---|---|
| *E. coli* | AAG-AAG | 244 | 0.08 | 0.08 | 1.01 |
| | AAG-AAA | 902 | 0.29 | 0.20 | 1.45 |
| | AAA-AAG | 544 | 0.18 | 0.20 | 0.87 |
| | AAA-AAA | 1416 | 0.46 | 0.52 | 0.88 |
| | AAG-AAG-AAG | 9 | 0.07 | 0.02 | 3.37 |
| | AAG-AAG-AAA | 20 | 0.16 | 0.06 | 2.89 |
| | AAG-AAA-AAG | 21 | 0.17 | 0.06 | 3.03 |
| | AAA-AAG-AAG | 4 | 0.03 | 0.06 | 0.58 |
| | AAG-AAA-AAA | 36 | 0.29 | 0.14 | 2.00 |
| | AAA-AAG-AAA | 29 | 0.23 | 0.14 | 1.61 |
| | AAA-AAA-AAG | 4 | 0.03 | 0.14 | 0.22 |
| | AAA-AAA-AAA | 1 | 0.01 | 0.38 | 0.02 |
| | AAG-AAG-AAA-AAG | 1 | 0.25 | 0.02 | 16.07 |
| | AAG-AAG-AAA-AAA | 1 | 0.25 | 0.04 | 6.20 |
| | AAA-AAG-AAA-AAA | 2 | 0.50 | 0.10 | 4.78 |
| *S. cerevisiae* | AAG-AAG | 3845 | 0.21 | 0.14 | 1.45 |
| | AAG-AAA | 5183 | 0.28 | 0.24 | 1.20 |
| | AAA-AAG | 4505 | 0.24 | 0.24 | 1.04 |
| | AAA-AAA | 4858 | 0.26 | 0.39 | 0.69 |
| | AAG-AAG-AAG | 261 | 0.16 | 0.05 | 2.87 |
| | AAG-AAG-AAA | 234 | 0.14 | 0.09 | 1.57 |
| | AAG-AAA-AAG | 224 | 0.13 | 0.09 | 1.51 |
| | AAA-AAG-AAG | 189 | 0.11 | 0.09 | 1.27 |
| | AAG-AAA-AAA | 211 | 0.13 | 0.15 | 0.87 |
| | AAA-AAG-AAA | 261 | 0.16 | 0.15 | 1.07 |
| | AAA-AAA-AAG | 117 | 0.07 | 0.15 | 0.48 |
| | AAA-AAA-AAA | 171 | 0.10 | 0.24 | 0.43 |
| | AAG-AAG-AAG-AAG | 24 | 0.10 | 0.02 | 4.88 |
| | AAA-AAG-AAG-AAG | 28 | 0.12 | 0.03 | 3.48 |
| | AAG-AAA-AAG-AAG | 23 | 0.10 | 0.03 | 2.86 |
| | AAG-AAG-AAA-AAG | 19 | 0.08 | 0.03 | 2.36 |
| | AAG-AAG-AAG-AAA | 27 | 0.11 | 0.03 | 3.35 |
| | AAG-AAG-AAA-AAA | 13 | 0.05 | 0.06 | 0.99 |
| | AAG-AAA-AAG-AAA | 19 | 0.08 | 0.06 | 1.44 |
| | AAA-AAG-AAG-AAA | 11 | 0.05 | 0.06 | 0.83 |
| | AAA-AAG-AAA-AAG | 17 | 0.07 | 0.06 | 1.29 |
| | AAG-AAA-AAA-AAG | 5 | 0.02 | 0.06 | 0.38 |
| | AAA-AAA-AAG-AAG | 9 | 0.04 | 0.06 | 0.68 |
| | AAG-AAA-AAA-AAA | 9 | 0.04 | 0.09 | 0.42 |
| | AAA-AAG-AAA-AAA | 14 | 0.06 | 0.09 | 0.65 |
| | AAA-AAA-AAG-AAA | 6 | 0.03 | 0.09 | 0.28 |

*Table 1. Continued on next page*

Table 1. Continued

| Organism | Sequence | Occurances | Fraction observed | Fraction expected | Enrichment |
|---|---|---|---|---|---|
| | AAA-AAA-AAA-AAG | 5 | 0.02 | 0.09 | 0.23 |
| | AAA-AAA-AAA-AAA | 9 | 0.04 | 0.15 | 0.25 |

The prevalence precise sequences encoding 2–3 consecutive lysine residues in *E. coli* and *S. cerevisiae* are displayed. The raw number of 'occurrences' are listed for each sequence. The enrichment values listed reflect the fraction observed/fraction expected.

factors involved in the degradation of RNA or proteins that might obscure interesting effects on translation. When the mCherry reporters (described in *Figure 1A*) were expressed in this system, we find that iterated AAA-containing reporters produce less protein than their iterated AAG-containing counterparts (*Figure 4A*, lanes 3 vs 4). Additionally, we note the appearance of a truncated protein product generated from the iterated AAA-containing reporter (*Figure 4A*, lane 3). This band is slightly larger than the size of protein produced when a stop-codon is positioned at the insertion site (*Figure 4A*, lanes 2–3).

To ask whether the truncated band is the typical product of a stalled ribosome, a peptidyl-tRNA, we subjected the products of our PURE reactions to RNase A treatment (*Figure 4B*). As a positive control, we observed that peptidyl-tRNA product generated from a non-stop mRNA (*Figure 4B*, lanes 9–10) does indeed change in mobility when treated with RNase A (see uppermost band resolve into smaller peptide products from this inefficiently translated mRNA). By contrast, the truncated band generated from the $(AAA)_{12}$-containing reporter does not shift in mobility on a gel following RNase A treatment (*Figure 4A*, lanes 5–6). We closely examined our reporter sequence and found that there are several out of frame stop-codons following the $(AAA)_{12}$ insert (*Supplementary file 1*). We next showed that the truncated band is generated by RF-mediated peptide release, likely on a canonical stop codon reached following ribosome sliding on poly(A) sequence (*Figure 4A*, lanes 7–8). Further experiments indicate that both RF1 and RF2 can promote release of this product and that the release reaction is independent of RF3 (*Figure 4—figure supplement 1*). The formation of truncated product from our $(AAA)_{12}$ reporters is a signature that reports on ribosome sliding on iterated AAA sequences. We note that the truncated band is also observed when the mCherry reporter is expressed in *E. coli* (and a western is performed with an α-HA antibody) (*Figure 4—figure supplement 2*). Together, these data provide evidence that ribosome slipping on iterated AAA sequences occurs both in a fully reconstituted translation system and in *E. coli*.

## Efficiency of ribosome sliding is dictated by consecutive A residues in the mRNA

To determine the minimum number of consecutive lysine or adenosine residues necessary for ribosomes to robustly slide on the iterated AAA-containing reporters, we expressed reporter constructs containing 3, 6, 9 or 12 lysines (encoded by AAA) in the PURExpress *E. coli* cell-free translation system (*Figure 5*). Truncated product (which we have determined to be a signature of ribosome sliding) was generated with as few as three consecutive lysines. We next asked whether the number of lysines residues or the number of consecutive adenosine nucleotides determines the extent of ribosome sliding. In this case, reporters were created containing a three lysine $(K_3)$ insert encoded by 9, 10, 11, or 13 As in a row (*Figure 5*). We find that an $A_{11}$ repeat results in the robust formation of truncated product (*Figure 5*, *Figure 5—figure supplement 1*) while little product is seen with $A_9$ or $A_{10}$ sequences, though each sequence encodes the same number of consecutive lysines.

## Poly(lysine) inserts that promote ribosome sliding are targeted by NMD in *S. cerevisiae*

In eukaryotic systems, NMD is a quality control system that recognizes mRNAs containing premature termination codons (PTC) and targets them for degradation. Upf1 is a key protein in NMD and *upf1Δ* cells stabilize PTC-containing transcripts. Previous studies established that when ribosomes frameshift during translation, these mRNAs are typically targeted for decay by NMD because the ribosomes

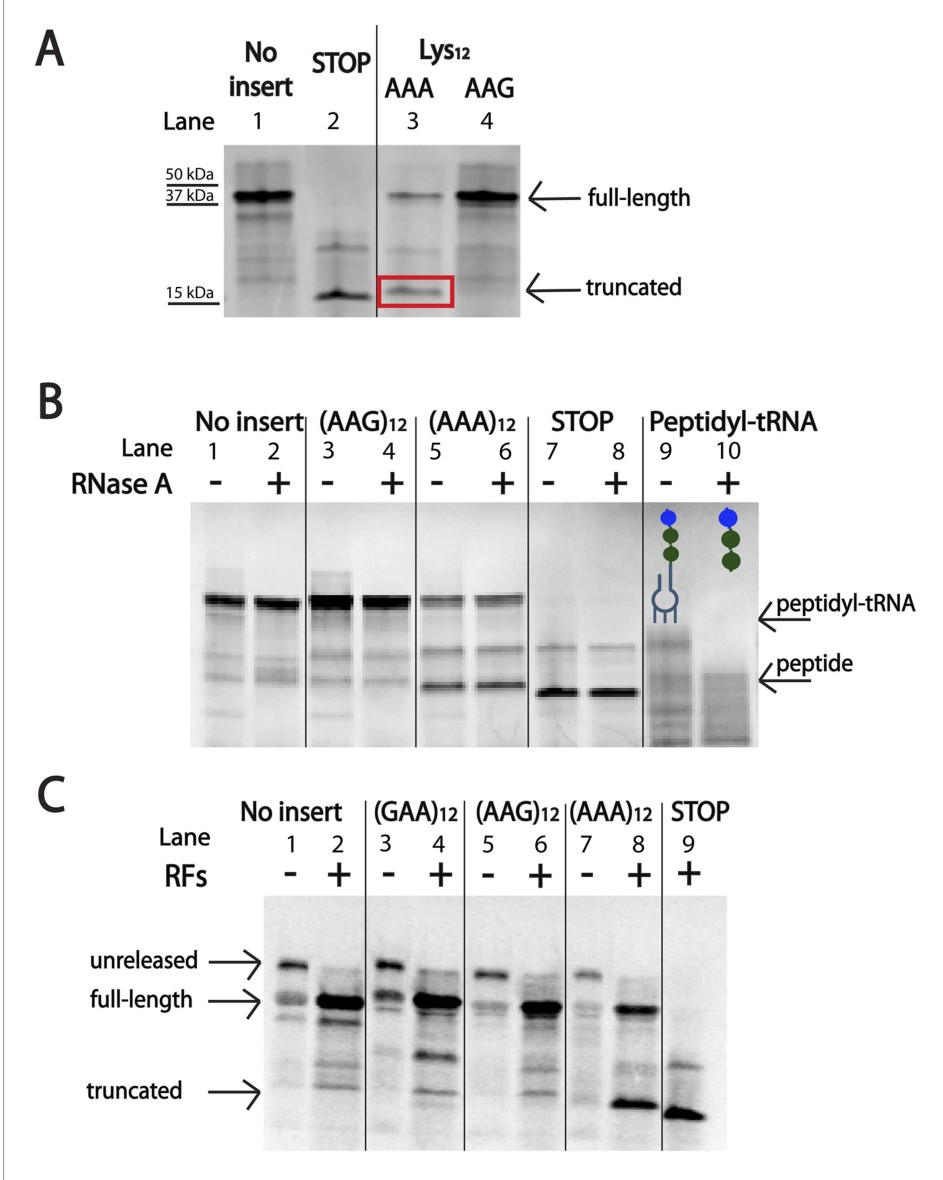

**Figure 4.** Ribosomes 'slide' into new frame on poly(A)-containing messages in the PURE in vitro translation system. (**A**) Expression of mCherry reporters (*Figure 1A*) in the *E. coli* PURE cell-free translation system (NEB). The truncated band generated from the (AAA)₁₂ reporter is boxed in red. The expected sizes of the full-length, STOP protein and truncated reporter are 42 kDa, 15 kDa, and 17 kDa, respectively. (**B**) Expression of mCherry reporters in the PURE system and subsequent treatment of peptide products with RNase A. Only the positive control (with a truncated mRNA species) yielded a peptidyl-tRNA product that shifted in mobility upon RNase A treatment. (**C**) Expression of mCherry reporters (*Figure 1A*) in the PURE in vitro translation system in the presence and absence of RFs (RFs = RF1, RF2, and RF3).

The following figure supplements are available for figure 4:

**Figure supplement 1**. Truncated product release is independent of RF3 in the PURExpress cell-free translation system.

**Figure supplement 2**. Western blot (α-HA) of mCherry reporters (*Figure 1A*) expressed in *E. coli*.

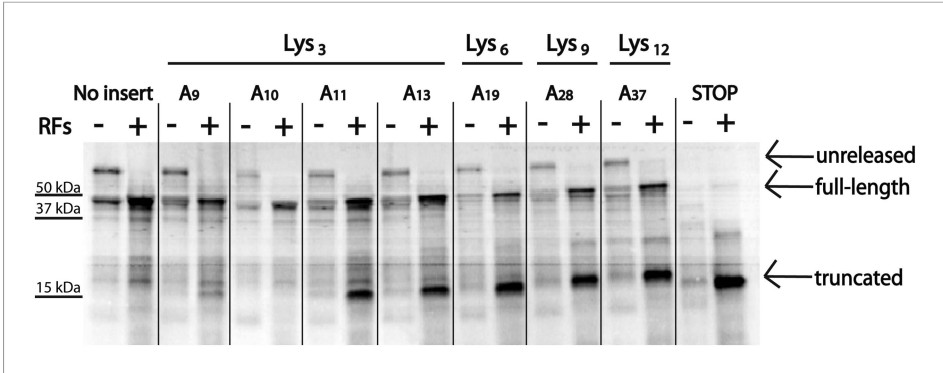

**Figure 5**. Position and length of poly(A) stretch contributes to ribosome 'sliding' in the PURE in vitro translation system. Expression of mCherry reporters containing poly(A) inserts of various lengths in the presence (+) and absence (−) of RFs.
The following figure supplement is available for figure 5:

**Figure supplement 1**. Quantification of the efficiency of ribosome sliding on mCherry reporters expressed in the PURExpress system.

generally encounter an out of frame premature termination codon (*Belew et al., 2011*, *2014*). We proposed that if the ribosome slides on iterated AAA-containing mRNAs in yeast, as it does in the bacterial system, then iterated AAA-containing mRNAs should be targeted by NMD. We addressed this possibility by measuring the levels of $(AAA)_{12}$, $(AAG)_{12}$, and $(AAGAAGAAA)_4$-containing mRNAs in two different yeast-expressed reporter systems (*Figure 1A*) in wild-type and *upf1Δ* cells.

First, as a control, we measured the mRNA levels of luciferase reporters containing no insert, an engineered premature stop codon (positive control), and a stem-loop known to trigger an alternative mRNA quality control pathway, no-go decay (negative control) (*Doma and Parker, 2006*). We find that the levels of mRNA for PTC and stem-loop containing reporters are lowered (PTC = 2 fold, stem-loop = 21 fold) relative to reporters with no insert in wild-type yeast cells. Moreover, as expected, the level of PTC, but not stem-loop-containing, mRNA is recovered when the reporters are expressed in *upf1Δ* cells (*Figure 6A*). When this same experiment was performed with a luciferase reporter containing an $(AAA)_{12}$ sequence, we find that reporter mRNA levels are substantially reduced in wild-type cells (>50-fold down), and that these levels are partially recovered in a *upf1Δ* strain (*Figure 6A*). These results suggest that the $(AAA)_{12}$-containing reporter is indeed a target of NMD in vivo.

To more directly compare our *S. cerevisiae* and *E. coli* results, we performed experiments instead using the related mCherry reporters (*Figure 1A*) with no insert, or a variety of lysine inserts (($AAG)_{12}$, $(AAA)_{12}$, and $(AAGAAGAAA)_4$). In addition to measuring the absolute levels of reporter mRNAs in wild-type and *upf1Δ* cells (*Figure 6*), we asked whether the rates of mRNA decay for these reporters are impacted in the *upf1Δ* knock-out background (*Figure 6B* and *Figure 6—figure supplement 1*). We chose to include a mixed AAA/AAG reporter in addition to the simpler AAA and AAG repeat reporters because this sequence is commonly used to report on the NSD phenomenon (*Dimitrova et al., 2009*; *Chiabudini et al., 2012*, *2014*). Indeed, a recent study with an $(AAGAAGAAA)_4$-containing reporter argued that a truncated product generated by such a construct resulted from an unusual release factor-dependent termination event on a sense (lysine) codon (*Chiabudini et al., 2014*). In an attempt to recapitulate these results, we directly looked for evidence of eRF1:eRF3-mediated termination activity on iterated lysine mRNAs in vitro using a yeast reconstituted translation system (*Shoemaker et al., 2010*); we see no evidence that such an event can occur (*Figure 6—figure supplement 2*). We propose that an alternative explanation for the published data could be that the ribosome slides out of frame on the $(AAGAAGAAA)_4$ sequence, resulting in premature termination on a previously out-of-frame stop codon, akin to what we observe in the PURE *E. coli* cell-free translation system (*Figure 4C*). This possibility seemed particularly likely given that we observed sliding activity on a AUG-AAA-AAG-UUC-STOP sequence in our in vitro reconstituted *E. coli* system (*Figure 3—figure supplement 1*).

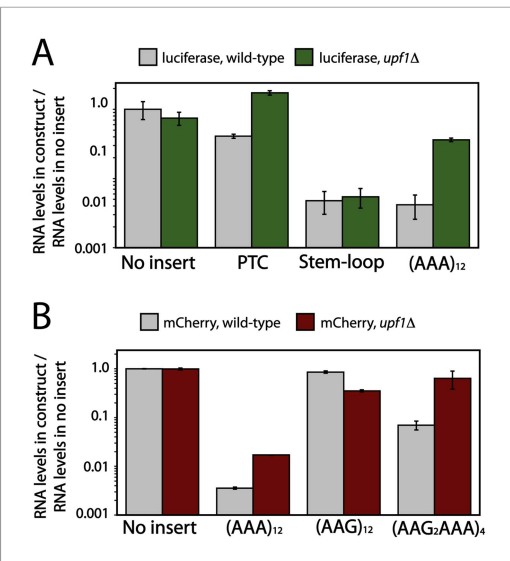

**Figure 6**. Deletion of Upf1p results in recovery of mRNA levels for poly(A) reporters in yeast. Luciferase (**A**) and mCherry (**B**) reporters (*Figure 1A*) were expressed in wild-type and *upf1Δ S. cerevisiae,* and the levels of reporter RNA were quantified by qRT-PCR. Various insertions including 12 lysines ((AAA)$_{12}$, (AAG)$_{12}$, (AAG$_2$AAA)$_4$), stem-loop, or premature termination codon (PTC) in the coding sequence are specified on the x-axes.

The following figure supplements are available for figure 6:

**Figure supplement 1**. mRNA half-life of reporter containing iterated AAA codons is Upf1 dependent.

**Figure supplement 2**. eRF1:3 does not prematurely terminate translation on coding sequences in poly (lysine) messages.

In wild type and *upf1Δ* cells, we find that the level of the (AAG)$_{12}$ containing reporter mRNA is unchanged relative to the mCherry reporter with no insert (*Figure 6B*). In contrast, the levels of (AAGAAGAAA)$_4$ and (AAA)$_{12}$ reporter mRNAs are significantly reduced compared to the control (no insert) reporter (15-fold and 30-fold, respectively). These observations are consistent with the low levels of protein expressed in vivo from these reporters relative to sequences containing no insert or (AAG)$_{12}$ (*Figure 1B*). As with the luciferase reporters, the level of mCherry mRNA containing an (AAA)$_{12}$ insert is partially recovered by the deletion of *UPF1* (*Figure 6B*). Strikingly, when the (AAGAAGAAA)$_4$-containing reporters are expressed in *upf1Δ* cells, the mRNA levels are nearly fully recovered. The mRNA half-lives for these reporters are similarly recovered in the *upf1Δ* cells (*Figure 6—figure supplement 1*). Thus both the (AAGAAGAAA)$_4$ and (AAA)$_{12}$ reporter mRNAs are targeted by NMD in yeast cells (*Figure 6B*). These results are consistent with a model invoking ribosome sliding followed by recognition of out-of-frame premature termination codons.

## Iterated AAA codons are selected against in yeast and bacteria coding regions

We performed bioinformatic analyses of fully annotated ORFs to evaluate the codon usage in sequences of consecutive lysines found in the *E. coli* and *S. cerevisiae* transcriptomes. In both organisms, AAA codons are found more commonly than AAG codons (62% AAA vs 38% AAG in yeast, and 72% AAA vs 28% AAG in bacteria); however, consecutive AAA codons are under-represented relative to their overall codon usage (*Table 1*). This is highlighted by the observation that the longer the stretch of lysines, the lower the likelihood of the motif being comprised solely of AAA codons (*Table 1*). Such an underrepresentation of AAA codons becomes pronounced in runs of 3 or 4 lysine codons in both organisms. In *E. coli*, only a single AAA-AAA-AAA sequence is present, which is 50-fold less common than expected based on the frequency of AAA codons; in contrast, (AAG)$_3$ sequences are found 3.3-fold more often than expected. In *S. cerevisiae*, the trends are similar; there are 2.3 and 4-fold fewer (AAA)$_3$ and (AAA)$_4$ sequences, respectively, than expected. Conversely, (AAG)$_3$ and (AAG)$_4$ sequences are threefold to fivefold more abundant than expected. These data together argue that evolution has selected against the use of long runs of A to encode sequential lysines within ORFs.

## Discussion

Although many of the major players in NSD have been identified, a high-resolution mechanistic understanding of how translation of poly(A) sequences triggers NSD has been missing. Here, we provide mechanistic insight into what initially happens when the ribosome encounters poly(A) sequence. First, we find that the expression of proteins containing poly(lysine) stretches is codon-dependent in both bacteria and eukaryotes, with reporters containing iterated AAA codons consistently producing less protein than those with equivalent AAG codons (*Figures 1, 4*).

This differential protein output is not the result of imprecise RNA polymerase action (*Figure 3—figure supplement 3*) nor likely of disparities in the rate of adding lysine codons (*Figure 2*); lysines are slowly incorporated on iterated AAA and AAG codons. Instead, the codon-dependent disparity primarily stems from an unusual sliding event that occurs when ribosomes encounter consecutive AAA codons (*Figures 3, 4*). Our observation that ribosomes can slide in multiple frames on iterated AAA sequences provides a rationale for consecutive AAA codons being substantially under-represented in open reading frames in most genomes (see Bioinformatic discussion below, *Table 1* and (unpublished data).

Our biochemical data in *E. coli* lead us to propose a model (*Figure 7*) for what happens to the ribosome during the translation of homopolymeric A sequences. On these messages, the first lysine is added quickly ($k_{1,obs}$) while subsequent lysines are added more slowly, causing the ribosome to pause. We note that the rate constants measured in the in vitro assay reflect all of the processes that can occur each time a new lysine moiety is added to the growing polypeptide chain (Lys-tRNA$^{Lys}$ binding, peptidyl-transfer, translocation, peptidyl-tRNA drop-off, 70S complex instability, etc). We suspect it to be unlikely that ribosome pausing is caused solely by dramatically large defects in peptidyl-transfer, but instead may result from ribosomes that become effectively inactivated (e.g. as a result of complex instability on homopolymeric A messages, etc). Whatever the cause for an initial ribosome pausing event on iterated AAA sequences, the ribosome can either slide or perform another round of peptide bond formation. If the ribosome slides such that another AAA codon is positioned in the A site, the next step will also be slow, while if sliding somehow positions a non-lysine codon in the A site, recovery from slow elongation may occur. In our in vitro system translating di-lysine messages, we are able to observe sliding when consecutive AAA-codons are present because we force a strong pause after MKK formation by leaving out downstream factors required for translation to proceed (*Figure 3*). Our data suggest that ribosome sliding on iterated AAA sequences is the major difference between the translation of poly(AAA)- and poly(AAG)-containing messages that results in substantially different protein outputs. While each sequential addition of lysine in an iterated AAG sequence may be slow, the ribosome maintains frame and ultimately is able to produce full-length protein. By contrast, with repeated AAA sequences, the ribosome can eventually escape the homopolymeric A sequence through repeated sliding events, often emerging out-of-frame from the A stretch, and thus unable to produce full-length protein.

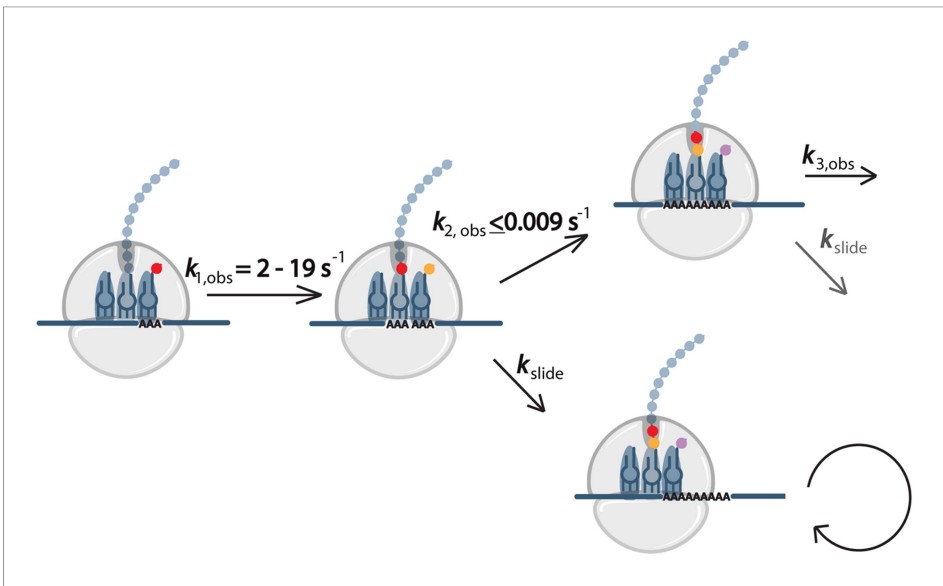

**Figure 7**. Model for events during ribosome sliding. In this model translation is paused following the addition of the first lysine. The ribosome can than either slide or perform another round of peptide bond formation. If an AAA codon is positioned in the A site after sliding, the next step will also be slow, while if sliding results in a non-lysine codon in the A site, recovery from slow elongation may occur.

Ribosome sliding on poly(A) is distinct from traditional programmed ribosomal movements such as +1 (*Farabaugh and Björk, 1999*; *Taliaferro and Farabaugh, 2007*) and −1 frame-shifts (*Dinman et al., 1991*; *Plant et al., 2003*; *Caliskan et al., 2014*; *Chen et al., 2014*; *Kim et al., 2014*). During a programmed frame-shifting (PRF) event, specific signals direct elongating ribosomes to shift reading frame by one base in the 5′ (−1) or 3′ (+1) direction (*Dinman, 2012*). −1 PRFs signals are typically characterized by a 'slippery' sequence (X XXY YYZ) that is modulated by the presence of a downstream secondary structure, most commonly a pseudoknot (*Plant et al., 2003*; *Jacobs et al., 2007*; *Caliskan et al., 2014*; *Chen et al., 2014*; *Kim et al., 2014*). The secondary structure impairs the normal movement of the ribosome during translocation, and promotes the frame-shift event in an EF-G dependent manner (*Caliskan et al., 2014*; *Chen et al., 2014*). +1 PRFs signals are more diverse than −1 PRFs, but still generally depend on a slippery sequence and a downstream element (e.g., secondary structure or rare codon) that causes the ribosome to pause (*Dinman, 2012*). Iterated A stretches are inherently slippery and contain a built-in translation pause (adding consecutive lysines is slow—*Figure 2*), however the poly(A) sequences that we have studied lack significant secondary structure downstream that might contribute to limiting unregulated ribosome sliding. As such, when ribosomes slide on iterated AAA codons, forward and backward movements may be permitted. The scale of the movements undergone during a ribosome sliding event may be more similar to those documented in translational bypassing on the gene product 60 of bacteriophage T4 which is synthesized from a discontinuous reading frame (*Samatova et al., 2014*). Importantly, however, in contrast to this specific concerted large-scale movement (50 nucleotides) which results in the production of a single peptide product, ribosome sliding is different in that no single outcome appears to be encoded by the event. The inability of the ribosome to translate a discrete product on homopolymeric A sequences likely explains the bioinformatic analyses demonstrating that poly(A) sequences are strongly selected against in coding sequences containing iterated lysines (*Table 1*). Consistent with this idea, in *E. coli* we find that the minimum length (11) of a homopolymeric A sequence needed to trigger ribosome sliding in the PURE cell-free translation system (*Figure 5*, and *Figure 5—figure supplement 1*) correlates with the length of lysine stretch at which homopolymeric sequences are selected against in mRNA coding regions (*Table 1*).

There are multiple reports in the literature indicating that frame-shifted ribosomes can trigger NMD (*Belew et al., 2011*, *2014*). We find that mRNA levels for reporters containing $(AAA)_{12}$ or $(AAGAAGAAA)_4$, but not $(AAG)_{12}$ sequences, are reduced in a Upf1-dependent manner. These data are consistent with the idea that sliding on homopolymeric A stretches can eventually lead to ribosomes reaching out-of-frame premature termination codons (*Figure 6*). A recent report in the literature argued that translation of poly(lysine) stretches led to an unusual termination event on a sense codon (AAA or AAG) mediated by eRF3 (presumably in concert with its binding partner eRF1) (*Chiabudini et al., 2014*). These observations bring to mind premature termination events on sense codons documented in *E. coli* (*Zaher and Green, 2009*); this quality control system was proposed to increase the fidelity of translation by minimizing frame-shifting and eliminating errors made during tRNA selection. We note that the premature termination event that we previously documented in *E. coli* was highly dependent on RF3, while the termination event documented in *E. coli* in this manuscript at homopolymeric A sequences is RF3-independent (*Figure 4—figure supplement 1*). Given the clear evidence that we provide for ribosome sliding in the *E. coli* system and the inability to observe eRF1:eRF3-mediated peptide release on homopolymeric A programmed yeast ribosome complexes in vitro (*Figure 6—figure supplement 2*), we suggest that the most likely explanation for the eRF3-dependent truncated product generated in yeast cells on $(AAGAAGAAA)_4$–encoding reporters in Chiabudini et al. is the result of ribosome sliding and canonical recognition of downstream premature stop codons. We note that there are multiple out-of-frame stop codons following the $(AAGAAGAAA)_4$-repeat that could account for the observed products in Chiabudini et al. (*Chiabudini et al., 2014*).

We were intrigued by the observation that the $(AAGAAGAAA)_4$ reporter mRNA levels are more efficiently recovered than those of the $(AAA)_{12}$ reporter mRNA in a *UPF1*-deletion strain. We speculate that the more modest sliding within the (AAGAAGAAA)-repeats might be distinguished from the sliding on (AAA)-repeats in an important way. Sliding within homopolymeric AAA sequence most typically results in another nearby AAA codon being poised in the A site, and another inefficient elongation event with Lys-tRNA$^{Lys}$. Ribosomes that eventually exit the poly(A) sequence to reach heteropolymeric sequence and an out-of-frame downstream premature stop codon will trigger NMD;

ribosomes that struggle to get past the very long stretch of iterated lysine codons will instead trigger NSD. As such, the mRNA levels for the (AAA)$_{12}$ reporter are partially recovered by a *UPF1* deletion and partially recovered by a *DOM34* deletion (data not shown). By contrast, on the (AAGAAGAAA)-repeat reporters, sliding has the potential to quickly place the ribosomes in a more productive frame for efficient elongation (one frame will result in Arg-Arg-Lys (RRK) repeats while the other frame will result in Glu-Glu-Lys (EEK) repeats). While we might predict that the poly(basic) RRKRRKRRKRR peptide will also be slowly translated, a ribosome that slides into the frame encoding the EEKEEKEEKEE peptide should be able to resume efficient elongation. As such, fewer ribosomes may trigger NSD and, instead, a majority of ribosomes will reach downstream premature stop codons that trigger NMD. These ideas can easily be understood in the context of the model in *Figure 7* where differences in the elongation rates (e.g. slow for iterated lysine residues but fast for incorporation of other amino acids) will impact the relative contribution of ribosome sliding to overall outcome.

NSD was originally identified by following the degradation of transcripts lacking termination codons (*Frischmeyer et al., 2002*; *van Hoof et al., 2002*). These studies led to the idea that NSD is triggered when the ribosome stalls while translating a poly(basic) lysine sequence. NSD is commonly studied using reporters in yeast that contain poly(basic) inserts; common lysine and arginine inserts that have been investigated include (AAA)$_{12}$, (AAG)$_{12}$, (AAG-AAG-AAA)$_4$, and (CGG-(CGA)$_2$-CGG-(CGC)$_2$)$_2$ (*Ito-Harashima et al., 2007*; *Dimitrova et al., 2009*; *Bengtson and Joazeiro, 2010*; *Brandman et al., 2012*; *Chiabudini et al., 2012*, *2014*). Consistent with our findings, previous studies reported differences in protein output in yeast when these different sequences are translated (*Ito-Harashima et al., 2007*; *Dimitrova et al., 2009*); iterated AAA codons are more detrimental to overall expression than iterated AAG codons. Despite these differences, because the mRNA and protein levels for all of these are broadly sensitive to known NSD factors (Ltn1, Dom34, Ski7), poly(basic) sequences have been treated equally. Our results demonstrating that ribosomes can slide on consecutive AAA codons suggest that there may be important distinctions to be made in considering these reporters and that there may be substantial mechanistic overlaps in these systems.

Even though cells rarely maintain homopolymeric A sequences in ORFs, there are some situations where the ribosome likely must deal with homopolymeric A stretches in both bacteria and eukaryotes. In bacteria, mRNAs are typically polyadenylated as part of the normal decay process (*Dreyfus and Régnier, 2002*). For example, ribosome sliding might provide an escape for ribosomes already engaged on these mRNAs (a form of ribosome rescue). In eukaryotes, virtually all mRNAs in the cell are polyadenylated, but usually a stop codon is found at the end of the encoded ORF. However, there is abundant recent evidence indicating that a significant portion of yeast (14%) and human (9%) genes contain at least one alternative polyadenylation site within their coding sequence (*Ozsolak et al., 2010*). It has even been suggested that premature polyadenylation may become up-regulated in cancerous cells (*Berg et al., 2012*). In cases where premature polyadenylation takes place within the ORF, the ribosome will surely encounter a homopolymeric A sequence, likely triggering so called Non-Stop-Decay (NSD). In light of the results presented here, we would suggest that the triggering of NSD (and associated mRNA decay, proteolysis and ribosome recycling) occurs following the slow translation of iterated lysines and ribosome sliding events. The ubiquity of premature polyadenylation suggests that NSD broadly serves as an important pathway for regulating gene expression. The observation of synonymous AAG to AAA changes in iterated lysine stretches in genes upregulated in cancer provides support for the significance of this mechanism of gene regulation (unpublished data). The widespread use of polyadenylation for non-coding purposes in mRNA transcripts may find its origins in the inability of the decoding machine, the ribosome, to carefully control the behavior of these sequences.

## Materials and methods

### Reporter creation

The Thrdx-HA-mCherry (*Figure 1A*, *Supplementary file 1*) no insert reporter expressed in *E. coli* and the PURExpress cell-free translation system was created using Gateway cloning to include the 2HA-mCherry sequence in the pBAD-DEST49 vector. The vectors containing inserts (Thrdx-HA-insert-mCherry: (AAA)$_{12}$, (AAA)$_6$, (AAG)$_{12}$, (AAGAAGAAA)$_4$, (GAA)$_{12}$, TAA (STOP), (A)$_{9-13}$, etc) were subsequently derived from this clone. To create the mCherry reporter expressed in yeast (*Figure 1A*), the Thrdx-HA-mCherry and Thrdx-HA-insert-mCherry sequences were amplified out of the pBAD-DEST49 vectors and cloned into the

p-ENTR/D-TOPO vector. The vector was then reacted with lr-clonease II to move the sequences into the pYES-DEST52 plasmid. The dual luciferase reporter described in *Figure 1A* was based on the dual luciferase plasmid from *Takacs et al. (2011)*. In this reporter, Renilla and Firefly luciferase are under the control of ADH and GPD promoters, respectively. We inserted sequences of interest into the N-terminus of *Renilla* luciferase. The single Renilla luciferase reporter described in *Figure 1A* was cloned into pYES2 (with a Gal promoter) using the Gateway cloning system.

## In vivo protein expression and visualization

Thrdx-HA-mCherry and Thrdx-HA-insert-mCherry constructs were expressed in 6 ml *E. coli* grown in LB-Ampicillin. The cells were grown to an OD of 0.4–0.6, induced with 25 µl of 5 g/10 ml arabinose, then harvested 2 hr post-induction. In yeast, the Thrdx-HA-mCherry constructs were expressed in wild-type and *upf1Δ S. cerevisiae* (BY4741) grown in 5 ml of–URA/+galactose media to an OD of 0.6. The single luciferase reporters were transformed into yeast and grown in–URA/+galactose media, and harvested at an OD of 0.6. Proteins production was analyzed via fluorescence, luminescence (*Figure 1*) or western blot analysis (*Figure 4—figure supplement 2*).

## Assessing lysine incorporation in fully reconstituted in vitro translation assays

70S initiation complexes (ICs) were prepared using *E. coli* ribosomes programmed with various mRNAs and f-[$^{35}$S]-Met-tRNA$^{Met}$ in the P site. mRNAs were generated by transcription with T7 polymerase and ICs were formed, pelleted, and resuspended as previously described (*Youngman et al., 2004*) on our messages of interest. Translation assays were initiated when equal volumes of ternary complex (10–20 µM charged tRNA, 12 µM EFG, 60 µM EfTu) were added to 0.2 nM 70S initiation complexes. Assays were performed in 219-Tris buffer (50 mM Tris pH 7.5, 70 mM NH$_4$Cl, 30 mM KCl, 7 mM MgCl$_2$, 5 mM βME). The limited addition of iterated lysines on a MK$_{A5}$-STOP message was also observed in polymix buffer (50 mM K$_2$HPO4 pH 7.5, 95 mM KCl, 5 mM NH$_4$Cl, 5 mM Mg(OAc)$_2$, 0.5 mM CaCl$_2$, 8 mM putrescine, 1 mM spermidine, 1 mM DTT). To measure the rates of amino acid incorporation, the reactions are quenched with 500 mM KOH (final concentration) at discrete time points (0 s–30 min) either by hand or on a quench-flow apparatus. For assays including release factors for the duration of the reaction (*Figure 3C*), RF1 and additional GTP were added prior to the initiation of translation (final concentrations 1 µM and 200 µM, respectively). The time-points were diluted 1:10 in nuclease free water and the reactants, intermediates and products visualized by electrophoretic TLC, as previously described (*Zaher and Green, 2009*). The reactants, products and intermediates were visualized by phosophorimaging and quantified with ImageQuant. The kinetic fits were modeled using Mathematica (details in *Figure 2—figure supplement 2*).

## Expression of reporters in the PURExpress in vitro translation system

The Thrdx-HA-mCherry and Thrdx-HA-insert-mCherry reporters were expressed in the PURExpress in vitro translation system (NEB, Ipswitch, MA) from PCR products. The peptidyl-tRNA construct was generated by creating a truncated mRNA lacking a stop codon directly after the Thrdx-HA sequence. The PURExpress reactions were initiated by mixing 1 µl of PCR product (29–22 ng/µl), 2 µl of solution A, 1.5 µl of solution B, and 0.6 µl of $^{35}$S-methionine. The reactions were run for 45–60 min at 37°C. Following translation, the products were immediately heat-denatured and loaded on a 4–12% Bis-Tris gel at 4°C in XT-MES buffer. For the experiments in which the PURExpress reaction products were treated with RNase A (*Figures 4B*), 0.5–1 µg of RNase A (Ambion, Grand Island, NY) was added to each reaction and solutions were incubated on ice for an additional 30 min before being denatured and loaded on a gel. The peptide products of the PURExpress reactions were visualized by Phosphoimager and quantified with ImageQuant (*Figure 3—figure supplement 2*, and *Figure 5—figure supplement 1*).

## Toeprinting assays

DNA templates were PCR amplified from plasmids (PCR-Blunt II-TOPO vector) encoding MEA(INSERT) EAEDYKDD sequences. The PURExpress cell-free transcription-translation system (NEB, Ipswich, MA) was used for in vitro protein synthesis. Reactions were run for 30 min at 37°C by mixing 0.2-pmol of DNA template, 2.5 µl of Solution A and 1 µl of Solution B along with either 0.5 µl of DMSO (5%) or thiostrepton (0.5 mm in 5% DMSO). 1 pmol of $^{32P}$ATP-labeled NV1 primer was added, and reverse transcription was performed with AMV as previously described (*Vazquez-Laslop et al., 2008*; *Tanner*

*et al., 2009*). Reactions were phenol and chloroform extracted, ethanol precipitated and visualized on a 6% denaturing PAGE gel. Sequencing lanes were generated from plasmids using the Sequenase 2.0 DNA sequencing kit (Affymetrix, Santa Clara, CA). All bands were visualized by PhosphorImager.

### Real-time quantitative reverse transcription PCR (qRT PCR) to measure reporter mRNA levels

*Reporter* mRNA levels were quantified by qRT-PCR using the iQ5 iCycler system (Bio-Rad, Hercules, CA) and iQ SYBR Green Supermix (Bio-Rad, Hercules, CA).

### Measuring mRNA decay

To measure the rate of mRNA decay in yeast for our mCherry reporters, we grew wild-type and *upf1Δ* cells expressing reporters in–ura/galactose media at 30°C to an OD600 of 0.4. Cells were washed three times with–ura media lacking sugar, then re-suspended in -ura/glucose media; the transcription of the reporter is shut-off by glucose. Samples were collected at discrete time points (0–90 min), and mRNA levels were analyzed by qRT PCR.

### Bioinformatic analyses

*E. coli* K-12 substrain MG1655 complete genome, 4140 ORFs (data source: GenBank:U00096.3; http://www.ncbi.nlm.nih.gov/nuccore/U00096.3) and *S. cerevisiae* 5887 verified ORFs (data source: http://downloads.yeastgenome.org/sequence/S288C_reference/orf_protein/) have been used for extraction of lysine codon numbers and analyses of consecutive codons shown in *Table 1*. Expected values for consecutive variants of lysine AAA and AAG codons were calculated based on observed values for a single AAA and AAG codons and their probabilities to be found in such arrangments. Observed values were calculated based on data from genomic distribution and total numbers of variants for two, three or four consecutive lys codons, respectively.

## Acknowledgements

We would like to thank Slavica Pavlovic-Djuranovic and Risa Burr for help with materials, Jon Lorsch for sharing the dual luciferase plasmid (*Takacs et al., 2011*), and Allen Buskirk for reading. We would also like to thank the National Institutes of Health (R37 GM059425 to RG, and F32 GM100608 to KSK) for funding and the Howard Hughes Medical Foundation (RG) for salary support.

## Additional information

### Competing interests

RG: Reviewing editor, *eLife.* The other authors declare that no competing interests exist.

### Funding

| Funder | Grant reference | Author |
|---|---|---|
| National Institute of General Medical Sciences (NIGMS) | F32 GM100608 | Kristin S Koutmou |
| Howard Hughes Medical Institute (HHMI) | Molcular Mechanisms of Translation and Their Implications for Gene Regulation | Rachel Green |
| National Institute of General Medical Sciences (NIGMS) | R37 GM059425 | Rachel Green |

The funders had no role in study design, data collection and interpretation, or the decision to submit the work for publication.

### Author contributions

KSK, SD, Conception and design, Acquisition of data, Analysis and interpretation of data, Drafting or revising the article; APS, Acquisition of data, Analysis and interpretation of data, Drafting or revising the article; JLB, Acquisition of data, Analysis and interpretation of data; AR, Analysis and interpretation of data, Drafting or revising the article; RG, Conception and design, Analysis and interpretation of data, Drafting or revising the article

# Additional files

**Supplementary file**
- Supplementary file 1. Primary sequence of mCherry with out of frame stop-codons highlighted. The nucleotide sequence of the Thrdx-HA-mCherry reporters (*Figure 1A*) with all out of frame stop codons after the insertion site highlighted in yellow.

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
