## [Decision Letter]

Thank you for sending your work entitled “Ribosomes slide on lysine-encoding homopolymeric A stretches” for consideration at *eLife*. Your article has been favorably evaluated by James Manley (Senior editor) and 3 reviewers, one of whom is a member of our Board of Reviewing Editors.

All the individuals responsible for the peer review of your submission have agreed to reveal their identity: Nahum Sonenberg (Reviewing editor and peer reviewer), Daniel Wilson (peer reviewer), and Jonathan Dinman (peer reviewer).

The Reviewing editor and the other reviewers discussed their comments before we reached this decision, and the Reviewing editor has assembled the following comments to help you prepare a revised submission.

The three reviewers concur that the manuscript presents convincing data for a novel mechanism of ribosome movement (sliding) on iterated AAA codons. The paper constitutes a strong combination of in vitro work using an *E. coli* system, coupled with the use of in vivo reporters both in *E. coli* and yeast. The experiments are well conducted and interpreted and support the conclusions. The kinetics analysis portion of this work is particularly notable.

There is only one major concern that was raised:

Figure 6: typically, the conclusion that an mRNA is a substrate for NMD requires a transcriptional arrest time course experiment. Simply monitoring mRNA steady state abundance +/- Upf1 is not sufficient. This experiment would need to be performed.

Another point relates to the toe-print assay that is in the supplementary material. You demonstrated convincingly that consecutive AAA codons cause ribosome slippage using a number of different reporter systems in the main figures. However, they are all indirect. The only direct reporter that is employed by is the toe-print assay that is in the supplementary. This assay should be in the main text. It could be presented in a more appropriate manner showing the nucleotide sequence (together with the amino acid sequence) so that the reader can understand why the data supports ribosome slippage, and exactly which frame the ribosomes are in with the different constructs. You should perhaps indicate this using an accompanying schematic. You might also consider using an mRNA with a unique codon downstream of the polylysine sequence in different frames so that you can monitor read-through and change of frame by omitting the unique amino acid from the translation reaction to stall ribosomes on this codon if they make it through the polylysine stretch.

Reviewer #1

1) Figure 1 (*S. cerevisiae*): is the difference between the negative impact of AAA12 and AAG12 on luciferase output significant? In addition, Firefly is mentioned to serve as an internal control in this experiment. However, it is not clear whether the construct they are using contains an IRES.

2) Figure 2:. it would be nice to show synthesis of products containing more than 5 lysine residues from the AUG-(AAA)5-UAA template. Did the authors try other TLC systems to resolve the MK_4+_ spot?

3) The authors mentioned that in *E. coli* a single tRNA decodes both lysine codons. However, it is not specified whether the anticodon of this tRNA is complementary to AAA or AAG codons. This might be important for understanding the underlying mechanism of ribosomal sliding.

4) Figure 6: the log scale numbers on the Y-axis should be corrected. In addition, this figure shows a huge drop in stability of AAA12 luciferase mRNA in wild-type cells (∼30 fold). Thus, one would expect even a greater effect of AAA12 insertion on luciferase production (considering both mRNA degradation and translational pausing). However, in Figure 1 the difference in luciferase production directed by No-insert and AAA12 mRNAs is only ∼5 fold. Why?

5) To render conclusions more general, it would be nice to explore the possibility of ribosome sliding on iterated AAA codons in a mammalian system, for example, in a rabbit reticulocyte lysate.

Reviewer #2

1) Do the authors consider peptidyl-tRNA drop-off as an outcome during slippage. Was this monitored?

2) Given the relatively large number of polylysine stretches in proteins of *E. coli* and particularly of yeast (where there are 171 (AAA)3 and 9 (AAA)4 occurrences), it would seem appropriate to look at available profiling data to see whether there is evidence for slow transation or frame-slippage (and accumulation at PTCs?), especially given the authors expertise in this technique. This would be particularly interesting within the context of a *upf1*, *abce1* or *dom34* mutants?

3) In Figure 3, the authors see generation of MK_4_ peptides, even using MK2stop mRNAs. This suggests, as shown in Figure 7, that the ribosomes should slip back so as to be able to load the AAA codon in the A-site. This would mean that a peptidyl-tRNALys would be in position to basepair with the AUG start codon, raising the question as to whether the codon prior to the polylysine sequence influences the rate of slippage and therefore lysine incorporation for the *k*_*2*_ rate constant? Could this be easily tested using a MXK2-stop mRNA?

4) It would be appropriate to have molecular weight marker sizes indicated in Figures 4 and 5, with the expected sizes of the products indicated in the legends.

Reviewer #3

1) Figure 1: it is not obvious why a dual-luciferase reporter vector was used for these experiments, nor the rationale for why the insert is placed at the 5' end of the *Renilla* sequence. An explanation of the rationale behind the setup would be appreciated, especially given that both reporters are used to study NMD in yeast cells (Figure 6).

2) Figure 3: the descriptions of Figure 3 in the manuscript text do not match up with the figures themselves. Furthermore, we suggest that the components of Figure 3 ought to be reordered. Namely, the di-lysine schematic diagram (currently 3B) should be introduced first.

3) Figure 6: I am somewhat worried that the luciferase mRNA steady state abundance in wild-type cells is only half that of UPF-delta cells. Usually, mRNAs harboring in frame PTCs are very low abundance (e.g. <5%). This may be due to the fact that a dual luciferase reporter was used? However, in the end, the data are what they are, and this is not a deal-breaker.

---

## [Author Response]

*The Reviewing editor and the other reviewers discussed their comments before we reached this decision, and the Reviewing editor has assembled the following comments to help you prepare a revised submission*.

*The three reviewers concur that the manuscript presents convincing data for a novel mechanism of ribosome movement (sliding) on iterated AAA codons. The paper constitutes a strong combination of in vitro work using an* E. coli *system, coupled with the use of in vivo reporters both in* E. coli *and yeast. The experiments are well conducted and interpreted and support the conclusions. The kinetics analysis portion of this work is particularly notable*.

*There is only one major concern that was raised*:

Figure 6*: typically, the conclusion that an mRNA is a substrate for NMD requires a transcriptional arrest time course experiment. Simply monitoring mRNA steady state abundance +/- Upf1 is not sufficient. This experiment would need to be performed*.

Figure 6—figure supplement 1 and the corresponding Results section now include an mRNA half-life experiment following transcriptional shut-off. This experiment is described in the Materials and methods.

*Another point relates to the toe-print assay that is in the supplementary material. You demonstrated convincingly that consecutive AAA codons cause ribosome slippage using a number of different reporter systems in the main figures. However, they are all indirect. The only direct reporter that is employed by is the toe-print assay that is in the supplementary. This assay should be in the main text. It could be presented in a more appropriate manner showing the nucleotide sequence (together with the amino acid sequence) so that the reader can understand why the data supports ribosome slippage, and exactly which frame the ribosomes are in with the different constructs. You should perhaps indicate this using an accompanying schematic. You might also consider using an mRNA with a unique codon downstream of the polylysine sequence in different frames so that you can monitor read-through and change of frame by omitting the unique amino acid from the translation reaction to stall ribosomes on this codon if they make it through the polylysine stretch*.

We updated the toe-print assay figure to include the nucleotide sequences as suggested and have included it as Figure 1–figure supplement 1. We believe that the toeprint data should remain as a supplement (Figure 1–figure supplement 1) to the main text because the toe-prints do not directly demonstrate sliding, and are intended to function simply in support of the data already in the main text. The addition of the toe-printing data would not add information beyond that already demonstrated. For example, the toe-prints indicate that ribosomes pause while adding sequential lysines, but our kinetic data more directly address this (Figure 2). Furthermore, while the toe-prints suggest that the ribosomes get out of frame on poly(A) messages, our PURE reactions in the presence and absence of release factors demonstrate quite clearly that ribosomes can get out of frame on homopolymeric A messages (Figures 4 and 5).

Reviewer #1

*1)*
Figure 1
*(*S. cerevisiae*): is the difference between the negative impact of AAA12 and AAG12 on luciferase output significant? In addition, Firefly is mentioned to serve as an internal control in this experiment. However, it is not clear whether the construct they are using contains an IRES*.

This questions posed here are now answered in the text. In brief, the difference between AAA_12_ and AAG_12_ on luciferase output in *S. cerevisiae* is significant (3-fold ± 1), and consistent with the differences we observed between mCherry expression with AAA_12_ and AAG_12_ inserts in *E. coli* (4-fold ± 0.3) (now in Results). Firefly does not contain an IRES but is under the control of a constitutively‐on GPD promoter (now in Materials and methods).

*2)*
Figure 2: *it would be nice to show synthesis of products containing more than 5 lysine residues from the AUG-(AAA)5-UAA template. Did the authors try other TLC systems to resolve the MK4+ spot?*

We tried a variety of TLC conditions (changing pH in the running buffer, time of run, etc.) in order to resolve the MK4+ spot. Unfortunately, we were unable to do so.

*3) The authors mentioned that in* E. coli *a single tRNA decodes both lysine codons. However, it is not specified whether the anticodon of this tRNA is complementary to AAA or AAG codons. This might be important for understanding the underlying mechanism of ribosomal sliding*.

The anticodon of this tRNA is complementary to AAA (and forms a wobble interaction therefore with AAG). This has been added to the text (Results).

*4)*
Figure 6*: the log scale numbers on the Y-axis should be corrected. In addition, this figure shows a huge drop in stability of AAA12 luciferase mRNA in wild-type cells (∼30 fold). Thus, one would expect even a greater effect of AAA12 insertion on luciferase production (considering both mRNA degradation and translational pausing). However, in*
Figure 1
*the difference in luciferase production directed by No-insert and AAA12 mRNAs is only ∼5 fold. Why?*

We chose to leave the log-scale on the y-axis of Figure 6 because this allows us to better show the wide range of values that we measure (ranging from 0.004 to 1). While we too were surprised by the lack of correlation between mRNA and protein expression levels, we propose that these steady state measurements may fail to capture tightly correlated measurements (i.e. because the protein half‐life is longer).

*5) To render conclusions more general, it would be nice to explore the possibility of ribosome sliding on iterated AAA codons in a mammalian system, for example, in a rabbit reticulocyte lysate*.

We have another manuscript submitted for publication (Arthur et al.) that investigates the expression of iterated AAA codons in human cells. The work in this paper suggests ribosome sliding also occurs in higher eukaryotes.

Reviewer #2

*1) Do the authors consider peptidyl-tRNA drop-off as an outcome during slippage*. *Was this monitored?*

We have looked for peptidyl-dropoff, and do not see strong evidence for this being a major contributor to the defects that we observe. Figure 4 illustrates the lack of a peptidyl-tRNA product from an iterated AAA reporter.

*2) Given the relatively large number of polylysine stretches in proteins of* E. coli *and particularly of yeast (where there are 171 (AAA)3 and 9 (AAA)4 occurrences), it would seem appropriate to look at available profiling data to see whether there is evidence for slow transation or frame-slippage (and accumulation at PTCs?), especially given the authors expertise in this technique. This would be particularly interesting within the context of a* upf1*,* abce1 *or* dom34 *mutants?*

The stretches of poly(A) found in ORFs in the genome in yeast and *E. coli* may not be long enough to trigger sliding *in vivo*. We would argue that nature has counter-selected such sequences that would promote sliding (Table 1), and we therefore we were not surprised that these were not evident in our profiling data (where we did indeed look). We would add that our profiling studies are inconclusive because of a potential technical limitation in the profiling; no foot-prints can be recovered from poly(A) stretches (unpublished data). We are currently trying to understand this technical bottleneck.

*3) In*
Figure 3*, the authors see generation of MK4 peptides, even using MK2stop mRNAs. This suggests, as shown in*
Figure 7, *that the ribosomes should slip back so as to be able to load the AAA codon in the A-site. This would mean that a peptidyl-tRNALys would be in position to basepair with the AUG start codon, raising the question as to whether the codon prior to the polylysine sequence influences the rate of slippage and therefore lysine incorporation for the* k_2_
*rate constant? Could this be easily tested using a MXK2-stop mRNA?*

We agree that this could be tested. However, it seems most likely that the peptidyl-tRNA just iteratively slides three nucleotides backwards, reiterating the final lysine codon, such that the peptidyl-tRNA remains constantly poised on an AAA codon in the P site.

*4) It would be appropriate to have molecular weight marker sizes indicated in*
Figures 4 and 5*, with the expected sizes of the products indicated in the legends*.

Figures 4 and 5 and their corresponding legends were updated as suggested by the reviewer.

Reviewer #3

*1)*
Figure 1*: it is not obvious why a dual-luciferase reporter vector was used for these experiments, nor the rationale for why the insert is placed at the 5' end of the* Renilla *sequence. An explanation of the rationale behind the setup would be appreciated, especially given that both reporters are used to study NMD in yeast cells (*Figure 6*).*

The dual luciferase reporter was used because we had these data from a previous study that we were conducting in the lab. We also conducted the study in Figure 1 with mCherry reporters in yeast. The levels of fluorescence from these latter constructs were too small to measure, though the data analyzed by western are consistent with the luciferase reporter data.

*2)*
Figure 3*: the descriptions of*
Figure 3
*in the manuscript text do not match up with the figures themselves. Furthermore, we suggest that the components of*
Figure 3
*ought to be reordered. Namely, the di-lysine schematic diagram (currently 3B) should be introduced first*.

The figure has been rearranged as suggested, and is now correctly referenced in the text.

*3)*
Figure 6*: I am somewhat worried that the luciferase mRNA steady state abundance in wild-type cells is only half that of UPF-delta cells. Usually, mRNAs harboring in frame PTCs are very low abundance (e.g. <5%). This may be due to the fact that a dual luciferase reporter was used? However, in the end, the data are what they are, and this is not a deal-breaker*.

The mRNA levels of the wild-type luciferase reporter in the mRNA steady-state assay do not differ when the reporter is expressed in wild-type and *upf1*Δ cells, rather these values are within error of each other (amount of luciferase mRNA in cell-line/luciferase mRNA in wt cells = 1 ± 0.28, for wt, and 0.75 ± 0.16, for *upf1*Δ).